rsos.royalsocietypublishing.org

chemical biology

microbial lipid, solid-state fermentation, mulberry branches, oleaginous fungi, lignocellulose

**Author for correspondence:**
Weichuan Qiao
e-mail: hgqwc@njfu.edu.cn

This article has been edited by the Royal Society of Chemistry, including the commissioning, peer review process and editorial aspects up to the point of acceptance.

# Microbial oil production from solid-state fermentation by a newly isolated oleaginous fungus, *Mucor circinelloides* Q531 from mulberry branches

Weichuan Qiao, Junqi Tao, Yang Luo, Tianhao Tang, Jiahui Miao and Qiwen Yang

Department of Environmental Engineering, Nanjing Forestry University, Nanjing, Jiangsu Province 210037, People's Republic of China

(iD) WQ, 0000-0001-9721-9364

In this study, a newly isolated oleaginous fungus, *Mucor circinelloides* (*M. circinelloides*) Q531, was able to convert mulberry branches into lipids. The highest yield and the maximum lipid content produced by the fungal cells were $42.43 \pm 4.01$ mg per gram dry substrate (gds) and $28.8 \pm 2.85\%$, respectively. The main components of lignocellulosic biomass were gradually reduced during solid-state fermentation (SSF). Cellulose, hemicellulose and lignin were decreased from 45.11, 31.39 and 17.36% to 41.48, 28.71, and 15.1%, respectively. Gas chromatography analysis showed that the major compositions of the fermented products were palmitic acid (C16:0, 18.42%), palmitoleic acid (C16:1, 5.56%), stearic acid (C18:0, 5.87%), oleic acid (C18:1, 33.89%), linoleic acid (C18:2, 14.45%) and γ-linolenic acid (C18:3 n6, 22.53%) after 2 days of SSF. The fatty acid methyl esters contained unsaturated fatty acids with a ratio of 75.95%. The composition and content obtained in this study are more advantageous than those of many other biomass lipids. Meanwhile, the oleaginous fungus had a high cellulase activity of $1.39 \pm 0.09$ FPU gds$^{-1}$. The results indicate that the enzyme activity of the isolated fungus was capable of converting the cellulose and hemicelluloses to available sugar monomers which are beneficial for the production of lipids.

## 1. Introduction

Mulberry, an important economic crop, is planted widely in many different countries. China has the largest planted area of mulberry

rsos.royalsocietypublishing.org R. Soc. open sci. 5: 180551

with about 626 000 ha [1,2]. In addition to the fruit, other parts of the mulberry tree are also harvested. Mulberry leaves are used in sericulture, and abundant mulberry branches are by-products of this process. Although mulberry branches have long been used in Chinese medicine [3,4], most are harvested for firewood or result in agro-waste every year, which results in notable environmental problems [5]. Therefore, efficient use of excess mulberry branches remains an important challenge. Many studies have been focused on exploring functional materials from mulberry trees in the past few years. Lignocellulose in mulberry can be used to produce slow-release urea fertilizer [6], biochar [5], cellulose whiskers [7], natural fibre [8] and scrimber [9]. Also, pectin [2], antiviral flavonoids [4] and mulberroside A [10] can be directly extracted from the bark or xylem of mulberry branches.

As lignocellulose biomass, mulberry branches are abundant in hemicellulose, cellulose and lignin. Mulberry branches contain 54.3% cellulose and 28.9% lignin [5]. These soluble and insoluble carbohydrates are widely recognized as a promising feedstock for biofuel and bioenergy [11–13]. Recently, published studies reported that lignocellulose biomass could be bioconverted into lipids by oleaginous microorganisms that can accumulate more than 20% (w/w) of lipids of its total dry biomass weight [14]. The principal oleaginous microbial species are bacteria, yeasts, fungi and microalgae. Microbial lipids, namely single cell oil (SCO), are considered as a promising feedstock for biodiesel production [15]. Usually, biodiesel is produced from vegetable oils from oleaginous plants, but its development and application have been hindered by the high cost of the feedstock. Therefore, it is necessary to explore new raw materials that reduce the price of biodiesel without competing with food production [16]. SCO constitutes a promising alternative for producing biodiesel due to some advantages such as short producing period, little required labour and ease of scale-up [17]. However, the SCO source material is usually starch or glucose, which is still consumed as human food. Therefore, waste lignocellulose biomass has become the ideal SCO source material. Also, the use of mulberry branches to produce SCO is a useful way of disposing of them.

Typically, lignocelluloses are first hydrolysed by acid or alkali, and then, the lignocellulosic hydrolysates, which contain the mono-sugar and xylose, will be fermented into microbial lipids by submerged fermentation (SmF). However, during pretreatment, various inhibitors are generated, mainly weak acids, furan derivatives and phenolic compounds. Most of these inhibitors are toxic to microbes, hindering their cell growth and lipid accumulation. Moreover, the high costs of pretreatment and the SmF system have become the major obstacles for the wide application of biodiesel production. To overcome the shortcomings of SmF, solid-state fermentation (SSF) has been explored regarding cultivation of oleaginous fungus [18–20] because a solid medium better simulates the natural habitat of the fungi [21]. SSF has several advantages over SmF, which include lower wastewater output, reduced energy requirements, simpler fermentation media, easier aeration and reduced bacterial contamination [22]. Furthermore, SSF has also many advantages such as smaller bioreactor volume, no need of organic solvents (which generally confer some level of toxicity for the extract) and higher productivity [23,24]. Try et al. studied the production of γ-decalactones using SSF by Y. lipolytica W29, and the production amount reached the high concentration (5 g l$^{-1}$) [25]. Lin et al. investigated that Aspergillus oryzae A-4 yielded a lipid of 36.6 mg per gram dry substrate (gds), and a cellulase activity of 1.82 FPU gds$^{-1}$ with 25.25% of holocellulose use in the substrates was detected on the 6th day in SSF of the wheat straw and bran mixture without pretreatment [19].

Mucor sp. is a natural fungal species, which can be used to produce phytase in SSF processes and has been successfully employed in the production of lipids in SmF processes [17,26]. Several species of this genus have also been proved to be able to accumulate lipids up to 30–60% of biomass [27–31]. However, research studies of Mucor sp. on producing microbial lipid by SSF have not been reported; moreover, lipid productivities of other species by SSF of biomass were still low [32,33]. In this study, we investigated the feasibility of direct bioconversion of mulberry branches into microbial lipids by using a newly isolated filamentous fungus that has been identified as Mucor circinelloides Q531 (M. circinelloides Q531). Furthermore, the characteristics of the microbial lipids from mulberry branches are studied to determine whether the lipids can be used as a feedstock for biodiesel.

# 2. Material and methods

## 2.1. Lignocellulosic material

Mulberry branches were obtained from Jiangsu province in China and air-dried for 5 days. The raw materials were ground into a size of 40 mesh, and then oven-dried at 80°C for 24 h before being stored in a dryer at room temperature.

rsos.royalsocietypublishing.org R. Soc. open sci. 5: 180551

## 2.2. Isolation and identification of the oleaginous fungi

The oleaginous fungi were isolated from rotten branches and leaves found on Purple Mountain, located in Nanjing, China. The collection of rotten material was inoculated onto potato dextrose agar medium (PDA medium, pH adjusted to 6.5) containing 0.0001% chloramphenicol, using the spread-plate and streak-plate techniques, and then incubated for 6 days at 28°C. Fungal strains were stained with the Sudan black B technique [34] to screen for fungi with high lipid content. The nine strains with high lipid content were identified as those with large lipid globules, isolated and purified at 28°C on PDA medium and stored at 4°C.

The selected oleaginous fungi were identified based on the 18S rDNA sequences of their internal transcribed spacer (ITS) regions [35]. Briefly, mycelia were harvested by filtration from a liquid culture after 2 days of growth and were transferred to sterile mortar before liquid nitrogen was added. Mycelia were ground into a fine powder. After being ground in the presence of liquid nitrogen, the genomic DNA was extracted using an Ezup column fungi genomic DNA purification kit (Sangon Biotech, Shanghai, China). The ITS region was amplified using the universal primers ITS1 (5′-TCCGTAGGTGAACCT GCGG-3′) and ITS4 (5′-TCCTCCGCTTATTGATATGC-3′). Polymerase chain reaction (PCR) amplification was performed by initial denaturation at 94°C for 3 min followed by 30 cycles of 94°C for 1 min, 55°C for 30 s and 72°C for 1 min, with a final extension at 72°C for 10 min. PCR fragments were purified using a SanPrep column DNA gel extraction kit (Sangon Biotech). The obtained sequences were BLAST-searched against the National Center for Biotechnology Information database. Closely related multiple sequences were aligned and corrected. The phylogenetic tree was constructed using the neighbour-joining program in MEGA 5.0 [36].

## 2.3. Solid-state fermentation

Three grams of dried lignocellulosic materials and 6 ml of mineral salt solution (MS solution contained $(NH_4)_2SO_4$, 1.7 g; $KH_2PO_4$, 2.0 g; $MgSO_4·7H_2O$, 0.5 g; $CaCl_2·2H_2O$, 0.2 g; $FeSO_4·7H_2O$, 0.01 g; $ZnSO_4·7H_2O$, 0.01 g; $MnSO_4·4H_2O$, 0.001 g; $CuSO_4·5H_2O$, 0.0005 g; 0.1% Tween-80 (w/v), pH adjusted to 6.5) were well mixed and added to culture dishes. After sterilization, the medium was cooled down and inoculated with 0.5 ml of a spore suspension containing $10^7$ spores gds$^{-1}$. The culture was incubated at 28°C, 70–80% humidity for 12 days in an artificial climate incubator. Lipid yield, cellulase activity, fungal biomass, residual sugars, component of biomass and fatty acid composition were detected at different times during SSF.

## 2.4. Analysis methods

### 2.4.1. Determination of lipid yield

The samples, which consisted of the whole fermented lignocellulosic material and fungal biomass, were washed twice with distilled water in order to remove the remaining salts, centrifuged again and freeze-dried to a constant weight for 48 h. Extraction of total lipids was performed according to the previous method [37] with the following modifications. The dry sample (0.5 g) was soaked with 6 ml of 4 M HCl in 50 ml centrifuge tubes for 30 min, transferred into boiling water for 20 min, and cooled instantly to −20°C for 10 min. After the first extraction, the sample was vortexed with 12 ml methanol/chloroform (2 : 1, v/v), the remaining cell lipids were further extracted once with 12 ml methanol/chloroform (1 : 1, v/v); and then twice with 12 ml methanol/chloroform (1 : 2, v/v) [38]. Each extraction step consisted of incubation for approximately 1 day at room temperature, after which they were centrifuged at 5000g for 5 min. The bottom organic phase was carefully sucked out through the pipette, then 6 ml 0.1% NaCl solution was added, followed by centrifugation at 5000g for 5 min. Finally, the chloroform phase was transferred to new centrifuge tubes by a pipette. The solvent was removed by the nitrogen purge method, and the total lipid content was measured gravimetrically. The lipid content of the cells in SSF was expressed as percentage of gram lipids per gram dry cell weight (%). The lipid yield in SSF was expressed as milligram lipids per gram dry substrate (mg gds$^{-1}$). The lipids in the raw material were subtracted from the total lipids extracted before calculation.

### 2.4.2. Determination of cellulase activity

To prepare the enzyme extract in SSF, the fermented residue was suspended in 100 ml citrate buffer (pH 4.8, 0.1 M), and the mixture was shaken at 37°C for 2 h at 180 r.p.m. before being centrifuged.

The clarified supernatant was used for measurement of the cellulase activity by filter paper assay according to the US National Renewable Energy Laboratory (NREL) [39]. The reactions were incubated at 50°C for 1 h, and the released reducing sugars were determined by the 3,5-dinitrosalicylic acid (DNS) method using glucose as the standard for cellulase activities [40]. One unit (U) of enzyme activity was defined as the amount of enzyme required to release 1 µmol of reducing sugar per minute under assay conditions. The enzymatic activity was expressed as filter paper unit per gram of dry substrate (FPU gds$^{-1}$).

### 2.4.3. Determination of fungal biomass

Fungal growth estimation in SSF was carried out by estimation of N-acetyl glucosamine released by the acid hydrolysis of chitin present in the cell wall of the fungi [26,41]. The obtained results were depicted as mg glucosamine per gram initial dry substrate (mg gds$^{-1}$).

### 2.4.4. Determination of cellulose, hemicellulose and lignin

For cellulose content assayed by means of an HNO$_3$-ethanol method, hemicellulose was measured according to a two-brominating method and lignin was determined by the 72% (w/w) H$_2$SO$_4$ method [42].

### 2.4.5. Determination of residual sugars

The residual sugars in the fermented residue were extracted by ultrapure water and were analysed using high performance liquid chromatography (HPLC) (Shimadzu LC-20A) equipped with a strong acid cation-exchange resin column (Bio-Rad Aminex HPX-87H) operating at 55°C and a differential refractive index detector (RID-10A) operating at 40°C. The mobile phase was dilute H$_2$SO$_4$ solution of 0.005 mol l$^{-1}$ at the flow rate of 0.6 ml min$^{-1}$.

### 2.4.6. Determination of fatty acid composition

Prior to GC analysis, the extracted lipid samples were transformed into their corresponding methyl esters by using the modified protocol [43]. Briefly, 0.4 M KOH-methanol reagent was added to the lipid samples at 65°C for 30 min, followed by treatment with 14% BF3-methanol reagent at 65°C for 10 min and extraction with n-hexane/saturated NaCl solution (2:1). The samples were centrifuged (3500$g$) and the supernatant was obtained. Fatty acid profiles of microbial oils were performed by a GC-2010 gas chromatograph (Shimadzu Inc., Kyoto, Japan) equipped with a cross-linked capillary Rtx-WAX column (30 m length, 0.32 mm internal diameter, 0.5 µm film thickness) and a flame ionization detector. The operating conditions were as follows: inlet temperature at 250°C, detector temperature at 280°C and initial oven temperature at 50°C held for 1 min then raised to 200°C at a rate of 25°C min$^{-1}$, continuous ramping of temperature to 230°C at a rate of 2°C min$^{-1}$, then held at 230°C for 23 min. Chromatographic peaks and retention times were identified by the comparison to a fatty acid methyl ester (FAME) standard mixture (Supelco 37-Component FAME Mix, Sigma-Aldrich, St. Louis, MO, USA), and individual peaks were quantified by means of external standards and their corresponding calibration curves.

# 3. Results and discussion

## 3.1. Screening and characterization of isolated fungi

In this study, nine types of different fungi were isolated. They were inoculated in SSF. After 6 days, the lipid content of these strains was detected (see electronic supplementary material, figure S1). One of the isolated fungi, Q531, which contained the highest lipid content, was selected for further study. The 18S rDNA sequence of the newly isolated strain was submitted to the GenBank under the accession number KU523400. The BLAST search results indicated that the sequence of this strain was found to share 99% max ident with those of *M. circinelloides*. The 18S rDNA sequence and its homologous sequences were analysed using MEGA 5.0 software and a phylogenetic tree was established, as shown in figure 1. The oleaginous fungi were stained with Sudan black B to detect the presence of blue-black lipid globules within the cells. Figure 2 shows lipid globules detected within the cells of *M. circinelloides* Q531.

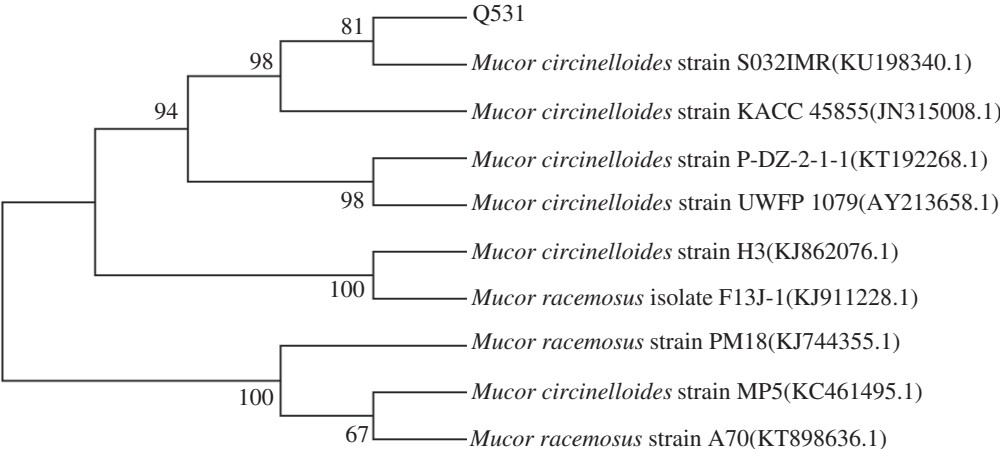

**Figure 1.** Neighbour-joining phylogenetic tree based on the 18S rDNA sequences shows the relationship between the newly isolated *M. circinelloides* Q531 and other members of the genus *Mucor*. The number of replicates is 1000. Numbers following the names of the strains are accession numbers of published sequences in the GenBank database.

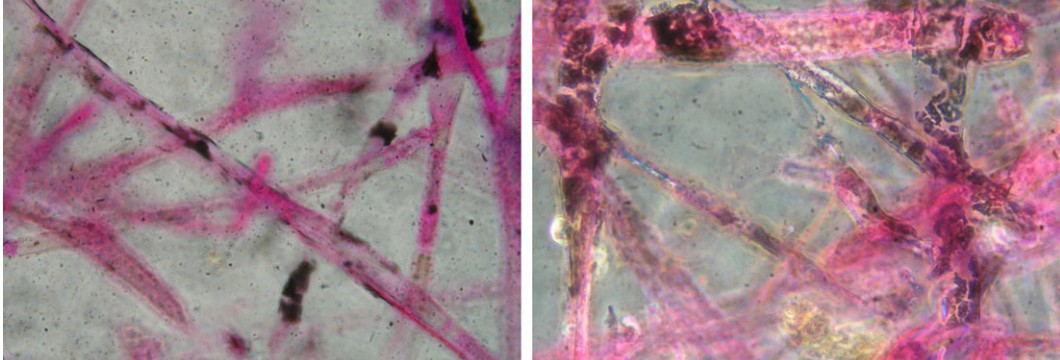

**Figure 2.** Photographs of lipid globules in the newly isolated *M. circinelloides* Q531 undergoing SSF taken by a microscope with a ×1000 oil immersion lens.

## 3.2. Lipid production and biomass by *M. circinelloides* Q531 in solid-state fermentation

As seen in figure 3, *M. circinelloides* Q531 has the key characteristics of a typical oleaginous species. Both the fungal biomass and lipid yield increased within 6 days, and the highest lipid yield was $42.43 \pm 4.01$ mg gds$^{-1}$. However, lipid yield decreased dramatically after day 6, while the biomass and lignocellulose use kept increasing, which might be attributed to the use of the storage lipids for biomass production. The biomass concentration drastically increased to the maximum of $134.56 \pm 1.41$ mg g$^{-1}$ and then started to decline slightly after day 8. The variation tendency of lipid content was similar to lipid yield in SSF. The lipid content of the fungal cells gradually increased to a maximum of $28.8 \pm 2.85\%$ after 6 days, as shown in tables 1 and 2. Compared with the previous investigations [32,44], the lipid content of this study is more than that of many other *Mucor* sp. that are used in active research. The results indicated that *M. circinelloides* Q531 was capable of converting widely available mulberry branches into microbial lipids.

## 3.3. The relationship between cellulase activity and lipid yield in solid-state fermentation

As shown in figure 3, the cellulase activity was rapidly increased (up to $1.39 \pm 0.09$ FPU gds$^{-1}$ after 6 days of SSF). Meanwhile, the production of lipids was also increased to $42.43 \pm 4.01$ mg gds$^{-1}$. However, the cellulase activity decreased dramatically following the decline of lipid yield after day 6 ($p < 0.05$), while the biomass kept increasing. As the direct conversion of lignocellulosic, biomass relies on the enzyme activity of the fungi to break down cellulose and hemicelluloses into available sugar monomers, including glucose, xylose and arabinose, for uptake, after which microorganisms use these sugars to produce lipid. Although a high carbon-to-nitrogen ratio is suitable for lipid accumulation, the enzyme activity is a crucial

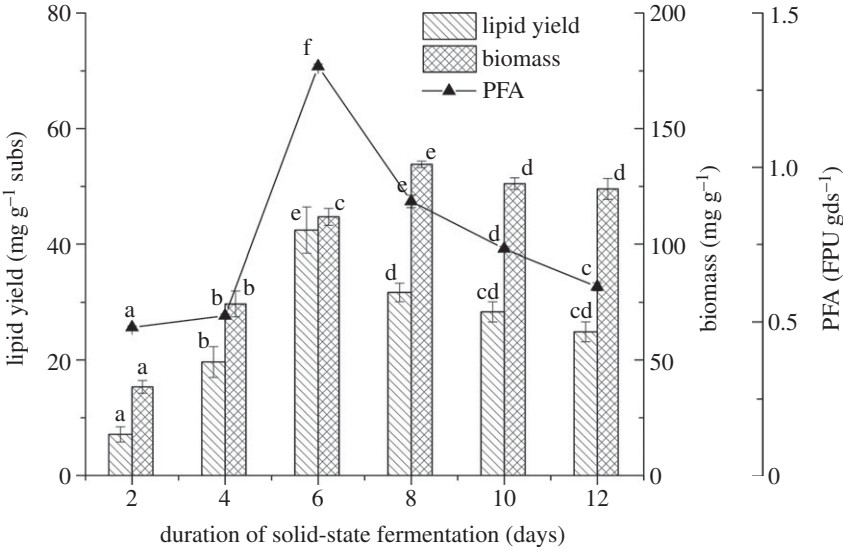

**Figure 3.** Lipid yield, biomass and cellulase activity of *M. circinelloides* Q531 in solid-state fermentation medium over time.

**Table 1.** The lipid content (%) by *M. circinelloides* Q531 through SSF over time.

| solid-state fermentation (days) | lipid content (%) |
|---|---|
| 2 | 10.95 $\pm$ 0.64 |
| 4 | 14.12 $\pm$ 1.79 |
| 6 | 28.8 $\pm$ 2.85 |
| 8 | 20.51 $\pm$ 0.93 |
| 10 | 17.09 $\pm$ 1.64 |
| 12 | 15.33 $\pm$ 1.28 |

**Table 2.** Weight and component loss of mulberry branches caused by *M. circinelloides* Q531 under SSF.

| SSF time (days) | weight loss (%) | component loss (%) | | |
|---|---|---|---|---|
| | | cellulose | hemicellulose | lignin |
| 6 | 12.45 $\pm$ 1.10 | 12.45 $\pm$ 1.10 | 16.14 $\pm$ 0.85 | 15.43 $\pm$ 0.42 |
| 12 | 18.37 $\pm$ 0.90 | 18.37 $\pm$ 0.90 | 25.79 $\pm$ 1.13 | 29.22 $\pm$ 1.11 |

factor for bioconversion of lignocellulosic biomass into lipids. Consequently, low cellulase activity would cause the limitation of available material in SSF and lead to the stored lipids being converted to new biomass. Similarly, Lin *et al.* [19] found that the lipid degradation did occur in SSF of *A. oryzae* A-4 in accordance with the decrease in its own cellulase activity. The results of this study indicate that *M. circinelloides* Q531 was capable of converting widely available lignocellulosic biomass of mulberry branches into microbial lipids. Moreover, high cellulose activity enhanced lipid production. It turned out that the committed step of lipid yield was microbial conversion of raw materials into sugar monomers by microorganisms in SSF. As shown in figure 4, the residual sugar concentration during SSF increased for 6 days, and then reduced, which was similar to the change of lipid product. The results showed that microbial production of lipid was accomplished by converting lignocellulose into sugar.

## 3.4. The composition of lignocellulose during the solid-state fermentation process

Before the beginning of SSF, the main content of lignocellulosic biomass (mulberry branches) consisted of cellulose, hemicellulose and lignin at 45.11, 31.39 and 17.36%, respectively. Cellulose, the main

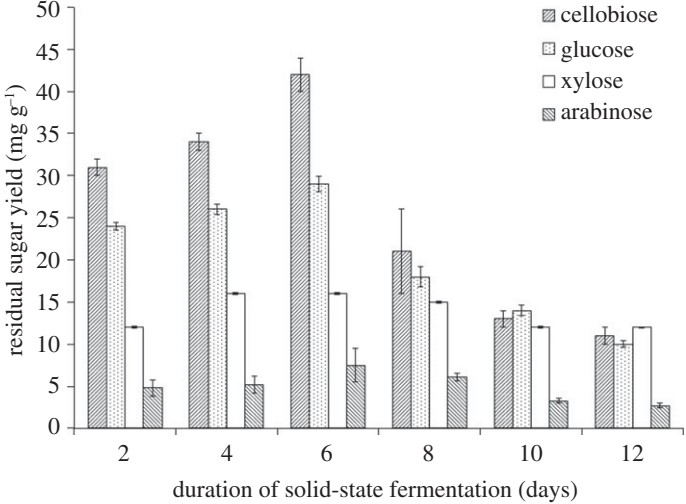

**Figure 4.** Residual sugars produced by *M. circinelloides* Q531 in solid-state fermentation medium over time.

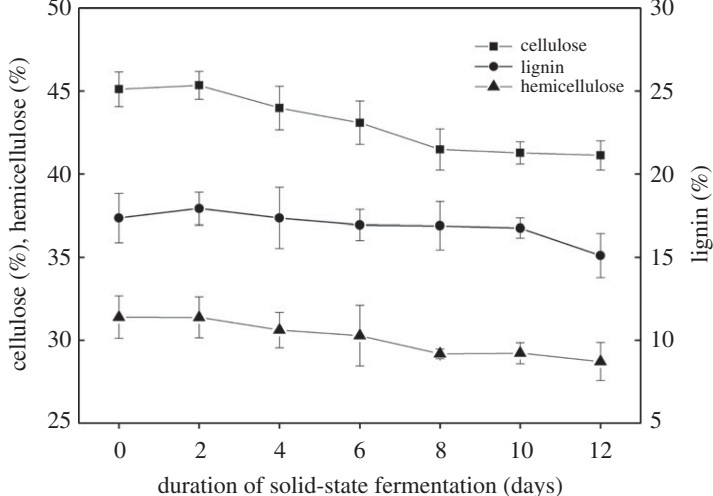

**Figure 5.** The change in lignocellulosic biomass (cellulose, hemicellulose and lignin) composition in solid-state fermentation over time.

fraction of plant cell walls, is linear and crystalline and is a homo-polymer of repeating units of glucose linked by $\beta$ (1−4) glycosidic bonds [45]; thus, the oleaginous fungi firstly use cellulose as a carbon source to grow. As seen in figure 5, the content of cellulose, in comparison to hemicellulose and lignin, was significantly decreased from 45.11% to 41.48% after 8 days. At the same time, the fungal biomass concentration rapidly increased to the maximum, but the lipid yield increased at first and then decreased, which might be attributed to the use of the storage lipids for biomass production (figure 3). Hemicellulose is a highly branched heteropolymer composed of D-xylose, D-arabinose, D-glucose, D-galactose and D-mannose [46], and its content gradually decreased from 31.39% to 28.71% in SSF for 12 days. Lignin, formed by polymerization of phenolic compounds, is hydrophobic in nature and is tightly bound to the cellulose and hemicellulose, protecting them from microbial [47] and chemical [48] degradation. It is hard to remove or modify the lignin (delignification) by microorganisms without different types of pretreatment methods. In figure 5, the results showed that the content of lignin started to decline observably after 10 days, and the content decreased from 17.36% to 15.1% in SSF. The content of cellulose, hemicellulose and lignin decreased by 5.92, 5.11 and 2.25%, respectively. This study indicated that *M. circinelloides* Q531 can use lignocellulosic biomass and change the chemical structure of mulberry branch in SSF. In addition, high degradation rate of cellulose and hemicellulose might benefit lipid production within a certain range of time.

rsos.royalsocietypublishing.org    R. Soc. open sci. 5: 180551

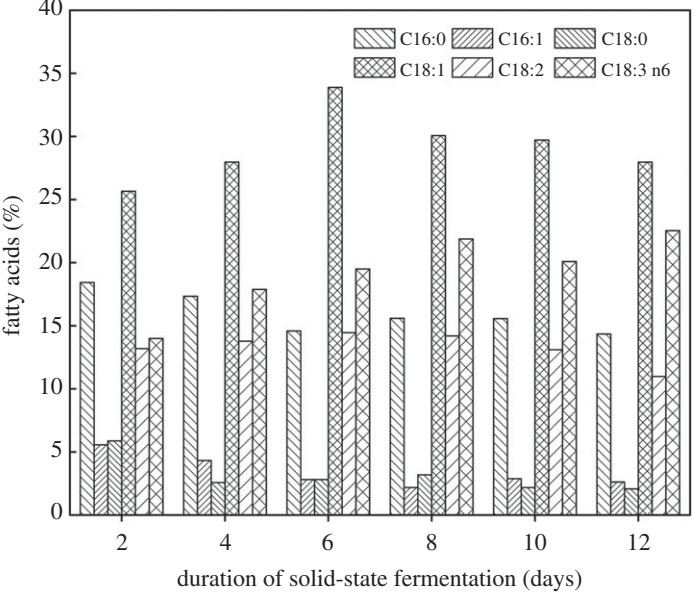

**Figure 6.** The composition of major fatty acids of lipids produced by *M. circinelloides* Q531 in solid-state fermentation over time.

**Table 3.** Fatty acid profile (%) of total lipids produced by *M. circinelloides* Q531 in solid-state fermentation of mulberry branches. Fatty acids including myristic (C14:0), palmitic (C16:0), palmitoleic (C16:1), stearic (C18:0), oleic (C18:1), linoleic (C18:2), γ-linolenic (C18:3 n6). SFA, saturated fatty acids; MUFA, mono-unsaturated fatty acids; PUFA, poly-unsaturated fatty acids.

| | solid-state fermentation (days) | | | | | |
|---|---|---|---|---|---|---|
| fatty acid type | 2 | 4 | 6 | 8 | 10 | 12 |
| C14:0 | 1.63 | 1.51 | 1.37 | 1.3 | 1.05 | 1.39 |
| C16:0 | 18.42 | 17.33 | 14.59 | 15.59 | 15.58 | 14.36 |
| C16:1 | 5.56 | 4.32 | 2.81 | 2.18 | 2.89 | 2.62 |
| C18:0 | 5.87 | 2.58 | 2.81 | 3.19 | 2.19 | 2.08 |
| C18:1 | 25.65 | 27.97 | 33.89 | 30.07 | 29.71 | 27.97 |
| C18:2 | 13.19 | 13.77 | 14.45 | 14.19 | 13.1 | 10.98 |
| C18:3 n6 | 14 | 17.88 | 19.51 | 21.87 | 20.09 | 22.53 |
| SFA | 33.76 | 28.74 | 24.06 | 25.89 | 26.52 | 26.87 |
| MUFA | 34.57 | 35.43 | 38.97 | 34.74 | 35.90 | 34.46 |
| PUFA | 31.67 | 35.83 | 36.98 | 39.38 | 37.59 | 38.67 |

## 3.5. GC analysis of FAMEs

The GC analysis results (figure 6) showed the major fatty acid compositions of total lipids from the SSF products at different fermentation time. After 2 days of SSF, the compositions were palmitic acid (C16:0, 18.42%), palmitoleic acid (C16:1, 5.56%), stearic acid (C18:0, 5.87%), oleic acid (C18:1, 33.89%), linoleic acid (C18:2, 14.45%) and γ-linolenic acid (C18:3 n6, 22.53%). During 12 days of SSF process, the content of five major fatty acids was continuously changed. The contents of palmitic acid (C16:0), palmitoleic acid (C16:1) and stearic acid (C18:0) were continuously reduced. However, the contents of oleic acid (C18:1) and linoleic acid (C18:2) reached a maximum of 33.89% and 14.45%, respectively, after 6 days. On the last day, the content of γ-linolenic acid (C18:3 n6) was 22.52%.

As revealed from the data (table 3), the total unsaturated fatty acid (USFA) content was 75.95% higher than saturated fatty acid (SFA, 24.06%) content. In addition, The USFA contained mono-unsaturated fatty

**Table 4.** Comparison of lipid and cellulase production by different oleaginous fungi grown on variety of inexpensive and non-edible agricultural and forestry residues in SSF.

| lignocellulosic biomass | strain | lipid yield (mg gds$^{-1}$) | cellulase | reference |
|---|---|---|---|---|
| wheat straw and bran | *Microsphaeropsis* sp. | 19–42 | 0.31–0.69 (FPU gds$^{-1}$) | [32] |
| wheat straw and bran | *A. oryzae* A-4 | 36.6 | 1.69 (FPU gds$^{-1}$) | [19] |
| rice straw and wheat bran | *Alternaria* sp. | 60.3 | 1.21 (FPU gds$^{-1}$) | [53] |
| rice straw | *M. elongate* PFY | 70.7 | —[a] | [54] |
| palm pressed fibre | *A. tubingensis* TSIP9 | 31.1 | <1.3 (U gds$^{-1}$) | [55] |
| palm empty fruit bunches | *A. tubingensis* TSIP9 | 37.5 | <1.3 (U gds$^{-1}$) | [55] |
| palm empty fruit bunch and palm kernel cake | *A. tubingensis* TSIP9 | 39.5 | 2.35 (U gds$^{-1}$) | [18] |
| mulberry branches | *M. circinelloides* Q531 | 42.4 | 1.39 (FPU gds$^{-1}$) | this study |

[a]Not available.

acid (MUFA, 38.97%) and poly-unsaturated fatty acid (PUFA, 36.98%). It is well known that high concentration of oleic acid (33.89%) in the obtained lipids was a desirable property for the production of biodiesel [49]. Moreover, γ-linolenic acid (22.53%), as a critical PUFA, was experimentally proved to have beneficial effects for the prevention and treatment of inflammatory disorders, diabetes, cardiovascular disorders, cancers and some other diseases [50]. Indeed, *M. circinelloides* was the first microorganism to be used commercially to produce an oil for human consumption—an oil rich in gamma-linoleic acid (GLA, 18:3; *cis*-6,9,12-octadecatrienoic acid) [51]. Similar fatty acid composition of lipids obtained from the oleaginous fungi was also found when *M. circinelloides* WJ11 was cultivated on K & R medium [52]. The fatty acid profile obtained in this study is similar to that of *M. circinelloides* CBS 203.28 cultivated on an acetic acid medium [33].

Lipid and cellulase production was compared in table 4 through SSF of lignocellulosic biomass by oleaginous fungi. As far as we know, the first attempt at direct lipid production from lignocellulosic biomass was by Peng *et al.* [32]. Their strains were able to directly produce lipids from a wheat straw and bran mixture with the lipid yield of 19–42 mg gds$^{-1}$. This was probably due to the cellulase activity of their strains (0.31–0.69 FPU gds$^{-1}$). Lin *et al.* [19] found that *A. oryzae* A-4 could produce the amount of lipid yield (36.6 mg gds$^{-1}$) by its own cellulose (1.69 FPU gds$^{-1}$). Dey *et al.* [53] discovered that *Alternaria* sp. was able to produce high cellulase activity of 1.21 FPU gds$^{-1}$ and lipid yield of 60.3 mg gds$^{-1}$. *M. elongate* PFY was another strain that has been reported for its lipid yield of 70.7 mg gds$^{-1}$ from rice straw but gave no available information of its enzyme activity [54]. *A. tubingensis* TSIP9 converted palm pressed fibre (PPF) and palm empty fruit bunches (EFB) into lipids with a maximum yield of 31.1 mg gds$^{-1}$ and 37.5 mg gds$^{-1}$, respectively; however, the strain produced very low cellulase activity (less than 1.3 U gds$^{-1}$) from both PPF and EFB [55]. Cheirsilp *et al.* [18] also found that *A. tubingensis* TSIP9 could use the mixture of palm empty fruit bunch and palm kernel cake to produce a lipid yield of 39.5 mg gds$^{-1}$, but the cellulose activity is very low (2.35 U gds$^{-1}$). In this study, it was clear that the cellulase activity and lipid yield of *M. circinelloides* Q531 were higher than the majority of those previously reported. It should be also noted that *M. circinelloides* Q531 is capable of converting lignocellulosic biomass into lipids by its own secretory cellulase. This study is the first to report that the newly isolated *M. circinelloides* directly converted lignocellulosic material into lipids with the highest yield, without exogenous cellulose and without pretreatment.

## 4. Conclusion

The newly isolated oleaginous fungus *M. circinelloides* Q531 exhibited satisfactory lipid production through SSF of mulberry branches. The fungi were able to use lignocellulosic biomass as the sole carbon and nutritional source. The main component of lignocellulosic biomass (including cellulose, hemicellulose and lignin) was continuously reduced during SSF. The maximum values of lipid yield, biomass, lipid content and cellulose activity were $42.43 \pm 4.01$ mg gds$^{-1}$, $134.56 \pm 1.41$ mg gds$^{-1}$,

rsos.royalsocietypublishing.org  R. Soc. open sci. **5**: 180551

$28.8 \pm 2.85\%$ and $1.39 \pm 0.09$ FPU gds$^{-1}$, respectively. In addition, the major fatty acids were palmitic acid (C16:0), oleic acid (C18:1), linoleic acid (C18:2) and γ-linolenic acid (C18:3 n6). *M. circinelloides* has the ability to greatly contribute toward the economics of biofuel production from cheap and abundant lignocellulosic biomass. Furthermore, the lipid, which was produced by *M. circinelloides* Q531 from mulberry branches through SSF, not only can be used as a biodiesel fuel but also can be processed into high-value-added products, such as GLA.

Ethics. All procedures performed in studies involving human participants were in accordance with the ethical standards of the institutional and/or national research committee and with the 1964 Helsinki declaration and its later amendments or comparable ethical standards. This article does not contain any studies with animals performed by any of the authors.

Data accessibility. DNA sequences: Genbank accessions KU523400.

Authors' contributions. T.T. carried out isolation and identification of the oleaginous fungi, participated in data analysis, carried out solid-state fermentation and determination of fungal biomass, participated in the design of the study and drafted the manuscript; J.M. carried out the acquisition of data, or analysis and interpretation of data; Q.Y. carried out determination of cellulose, hemicellulose and lignin, cellulose activity, lipid yield and fatty acid composition; W.Q. conceived of the study, designed the study, coordinated the study, helped draft the manuscript and revised it critically for important intellectual content, and gave final approval of the version to be published. All authors gave final approval for publication. All authors agreed to be accountable for all aspects of the work in ensuring that questions related to the accuracy or integrity of any part of the work are appropriately investigated and resolved.

Competing interests. We have no competing interests.

Funding. The study was supported by a project funded by the Priority Academic Program Development of Jiangsu Higher Education Institutions (PAPD).

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
