## [Reviewer comments · Royal Society Open Science]

Review History

RSOS-180551.R0 (Original submission)

Review form: Reviewer 1

Is the manuscript scientifically sound in its present form?

No

Are the interpretations and conclusions justified by the results?

Yes

Is the language acceptable?

Yes

Is it clear how to access all supporting data?

Not Applicable

Do you have any ethical concerns with this paper?

No

Have you any concerns about statistical analyses in this paper?

I do not feel qualified to assess the statistics

Recommendation?

Reject

Comments to the Author(s)

In this work, the authors isolated an oleaginous fungus, *Mucor circinelloides* Q531 and studied its lipid-producing ability and the composition of the lipid. However, the written English of this manuscript should be promoted. Most importantly, the novelty of this manuscript is insufficient. So many previous have screened several microbes with lipid-producing ability and the lipid produced by *M. circinelloides* Q531 is less than previous work (reference 51).

There are some comments as following:

1. The authors described that 'lipid yield decreased dramatically after d 6' in page 4, however, additional significance test can make this result being more persuasive.
2. Check throughout the manuscript and figures, all of the species names should be italic, and abbreviation of species name firstly occurred should be avoided.

Review form: Reviewer 2 (Xiang Li)

Is the manuscript scientifically sound in its present form?

Yes

Are the interpretations and conclusions justified by the results?

Yes

Is the language acceptable?

No

Is it clear how to access all supporting data?

Not Applicable

Do you have any ethical concerns with this paper?

No

Have you any concerns about statistical analyses in this paper?

Yes

Recommendation?

Accept with minor revision (please list in comments)

Comments to the Author(s)

The manuscript entitled " Microbial oil production from solid-state fermentation by newly isolated oleaginous fungi from mulberry branches" provided a newly isolated fungi that could convert mulberry branches to valuable product oil. The biomass and oil yield with cellulase activity, composition of the substrate and fatty acids were illustrated along with fermentation time. This paper need be carefully revised before being accepted in Royal Society Open Science.

Major problems:

1. Since the peak value of the liquid was achieved at d 6. Authors should focus the discussion during d 0 to d 6. From Fig 5, slightly decrease of cellulose (from 45% to ~43% at d 6) could be observed (estimate p value >0.05, if do ANOVA). Lignin almost unchanged (~37%) and Hemicellulose changed from 32% to 31% (d 6). However, yield of liquid increased significantly from 5-45 mg/gds. What is the real substrate for producing oil considering the carbon mass balance? The authors are suggested to provide the carbon mass balance analysis.
2. One page 4 line 37, since the authors use "significant" to express the decrease, ANOVA should be provided.

Minor problems:

1. Suggestion for Title: add the name of the fungi *Mucor circinelloides* to the title.
2. Suggestions for Summary: detail the time to obtain the GC analysis in line27: like at what day achieved palmitic acid (C16:0, 18.42%), palmitoleic acid (C16:1, 5.56%), stearic acid (C18:0, 5.87%), oleic acid (C18:1, 33.89%), et al. rephrase the sentence in line30 "Meanwhile, the oleaginous fungi have a high enzyme activity of..." Language and expressions should be polished carefully, rephrase the sentence in line23 : "The a high yield...reached by..."
3. Introduction part: the authors are suggested to emphasize the key innovative point in the introduction part, but the current manuscript is just a plain description of SSF and natural fungal species. What problems did the authors solve? And Why this is interesting to the readership? Like newly isolated? Why the productivity of this strain is higher than the literature?
4. Detail the screening process in chapter 3.2 from Materials and Methods, like what are the nine types? And what is the lipid content of the selected fungi from others? What is the total substrate for solid-state fermentation in 3.3 chapter? Also add one sentence that indicate which fermentative indexes assayed, including liquid yield, cellulase activity, fungal biomass, et al." Combine 3.5, 3.6 and 3.7 as one chapter.
5. Integrate Fig. 3 and Fig 4 as one Figure, which could be referred to the Fig.2 from one literature (Waste Management, 2018, 76, 414-422). Be uniform as mentioned in Table 1 and Table 2 that the fermentation time (days in Fig.3 to Fig. 6) should be changed to (d).
6. The GC data in line 55 on page 4 should be exhibited with the fermentation time. When did the compositions obtained?

Review form: Reviewer 3

Is the manuscript scientifically sound in its present form?

Yes

Are the interpretations and conclusions justified by the results?

No

Is the language acceptable?

Yes

Is it clear how to access all supporting data?

No

Do you have any ethical concerns with this paper?

No

Have you any concerns about statistical analyses in this paper?

No

Recommendation?

Major revision is needed (please make suggestions in comments)

Comments to the Author(s)

An isolated strain can directly use mulberry branches as materials to produce microbial lipid, which is an interesting works and dramatically saves the cost of fermentation. Besides, the writing of the manuscript is good. Attention, I think that the fourth suggestion is very important, you should make a serious consideration.

1 For the part of introduction, line 3 to 10, page 2, please supplement more information so that readers can understand how to resolve the shortcomings of SmF by SSF.

2 Line 47-49, page 2, the content can't be read. Revise it.

3 Part 4.2, no information showing the change of lignocellulose was observed. Please check it.

4 Part 4.3, Cellulase activity has indirect effect on the lipid yield. The change in sugars concentration (glucose, maltose, etc.) during the fermentation is better beneficial to interpret the increase or decline of lipid yield. So, supplement the experimental data.

Decision letter (RSOS-180551.R0)

20-Aug-2018

Dear Dr Qiao:

Title: Microbial oil production from solid-state fermentation by newly isolated oleaginous fungi from mulberry branches

Manuscript ID: RSOS-180551

The editor assigned to your manuscript has now received comments from reviewers. We would like you to revise your paper in accordance with the referee and Subject Editor suggestions which can be found below (not including confidential reports to the Editor). Please note this decision does not guarantee eventual acceptance.

Please submit your revised paper before 12-Sep-2018. Please note that the revision deadline will expire at 00.00am on this date. If we do not hear from you within this time then it will be assumed that the paper has been withdrawn. In exceptional circumstances, extensions may be possible if agreed with the Editorial Office in advance. We do not allow multiple rounds of revision so we urge you to make every effort to fully address all of the comments at this stage. If deemed necessary by the Editors, your manuscript will be sent back to one or more of the original reviewers for assessment. If the original reviewers are not available we may invite new reviewers.

When submitting your revised manuscript, you must respond to the comments made by the referees and upload a file "Response to Referees" in "Section 6 - File Upload". Please use this to

document how you have responded to the comments, and the adjustments you have made. In order to expedite the processing of the revised manuscript, please be as specific as possible in your response.

Yours sincerely,
 Dr Laura Smith, MRSC
 Publishing Editor, Journals
 Royal Society of Chemistry,
 Thomas Graham House,
 Science Park, Milton Road,
 Cambridge, CB4 0WF, UK

Royal Society Open Science - Chemistry Editorial Office

RSC Associate Editor:
 Comments to the Author:
 (There are no comments.)

RSC Subject Editor:
 Comments to the Author:
 (There are no comments.)

Reviewers' Comments to Author:
 Reviewer: 1

Comments to the Author(s)

In this work, the authors isolated an oleaginous fungus, *Mucor circinelloides* Q531 and studied its lipid-producing ability and the composition of the lipid. However, the written English of this manuscript should be promoted. Most importantly, the novelty of this manuscript is insufficient. So many previous have screened several microbes with lipid-producing ability and the lipid produced by *M. circinelloides* Q531 is less than previous work (reference 51).

There are some comments as following:

1. The authors described that 'lipid yield decreased dramatically after d 6' in page 4, however, additional significance test can make this result being more persuasive.
2. Check throughout the manuscript and figures, all of the species names should be italic, and abbreviation of species name firstly occurred should be avoided.

Reviewer: 2

Comments to the Author(s)

The manuscript entitled " Microbial oil production from solid-state fermentation by newly isolated oleaginous fungi from mulberry branches" provided a newly isolated fungi that could

convert mulberry branches to valuable product oil. The biomass and oil yield with cellulase activity, composition of the substrate and fatty acids were illustrated along with fermentation time. This paper need be carefully revised before being accepted in Royal Society Open Science. Major problems:

1. Since the peak value of the liquid was achieved at d 6. Authors should focus the discussion during d 0 to d 6. From Fig 5, slightly decrease of cellulose (from 45% to ~43% at d 6) could be observed (estimate p value >0.05, if do ANOVA). Lignin almost unchanged (~37%) and Hemicellulose changed from 32% to 31% (d 6). However, yield of liquid increased significantly from 5-45 mg/gds. What is the real substrate for producing oil considering the carbon mass balance? The authors are suggested to provide the carbon mass balance analysis.
2. One page 4 line 37, since the authors use "significant" to express the decrease, ANOVA should be provided.

Minor problems:

1. Suggestion for Title: add the name of the fungi *Mucor circinelloides* to the title.
2. Suggestions for Summary: detail the time to obtain the GC analysis in line27: like at what day achieved palmitic acid (C16:0, 18.42%), palmitoleic acid (C16:1, 5.56%), stearic acid (C18:0, 5.87%), oleic acid (C18:1, 33.89%), et al. rephrase the sentence in line30 "Meanwhile, the oleaginous fungi have a high enzyme activity of..." Language and expressions should be polished carefully, rephrase the sentence in line23 : "The a high yield...reached by...",
3. Introduction part: the authors are suggested to emphasize the key innovative point in the introduction part, but the current manuscript is just a plain description of SSF and natural fungal species. What problems did the authors solve? And Why this is interesting to the readership? Like newly isolated? Why the productivity of this strain is higher than the literature?
4. Detail the screening process in chapter 3.2 from Materials and Methods, like what are the nine types? And what is the lipid content of the selected fungi from others? What is the total substrate for solid-state fermentation in 3.3 chapter? Also add one sentence that indicate which fermentative indexes assayed, including liquid yield, cellulase activity, fungal biomass, et al." Combine 3.5, 3.6 and 3.7 as one chapter.
5. Integrate Fig. 3 and Fig 4 as one Figure, which could be referred to the Fig.2 from one literature (Waste Management, 2018, 76, 414-422). Be uniform as mentioned in Table 1 and Table 2 that the fermentation time (days in Fig.3 to Fig. 6) should be changed to (d).
6. The GC data in line 55 on page 4 should be exhibited with the fermentation time. When did the compositions obtained?

Reviewer: 3

Comments to the Author(s)

An isolated strain can directly use mulberry branches as materials to produce microbial lipid, which is an interesting works and dramatically saves the cost of fermentation. Besides, the writing of the manuscript is good. Attention, I think that the fourth suggestion is very important, you should make a serious consideration.

- 1 For the part of introduction, line 3 to 10, page 2, please supplement more information so that readers can understand how to resolve the shortcomings of SmF by SSF.
- 2 Line 47-49, page 2, the content can't be read. Revise it.
- 3 Part 4.2, no information showing the change of lignocellulose was observed. Please check it.
- 4 Part 4.3, Cellulase activity has indirect effect on the lipid yield. The change in sugars concentration (glucose, maltose, etc.) during the fermentation is better beneficial to interpret the increase or decline of lipid yield. So, supplement the experimental data.

Author's Response to Decision Letter for (RSOS-180551.R0)

See Appendix A.

RSOS-180551.R1 (Revision)

Review form: Reviewer 1

Is the manuscript scientifically sound in its present form?

Yes

Are the interpretations and conclusions justified by the results?

Yes

Is the language acceptable?

Yes

Is it clear how to access all supporting data?

Yes

Do you have any ethical concerns with this paper?

No

Have you any concerns about statistical analyses in this paper?

No

Recommendation?

Accept as is

Comments to the Author(s)

All my concerns have been addressed.

Review form: Reviewer 2 (Xiang Li)

Is the manuscript scientifically sound in its present form?

Yes

Are the interpretations and conclusions justified by the results?

Yes

Is the language acceptable?

Yes

Is it clear how to access all supporting data?

Yes

Do you have any ethical concerns with this paper?

No

Have you any concerns about statistical analyses in this paper?

No

Recommendation?

Accept as is

Comments to the Author(s)

The authors have addressed all the questions. The quality has been improved.

Decision letter (RSOS-180551.R1)

18-Oct-2018

Dear Dr Qiao:

Title: Microbial oil production from solid-state fermentation by newly isolated oleaginous fungi, *Mucor circinelloides* Q531 from mulberry branches
Manuscript ID: RSOS-180551.R1

It is a pleasure to accept your manuscript in its current form for publication in Royal Society Open Science. The chemistry content of Royal Society Open Science is published in collaboration with the Royal Society of Chemistry.

RSC Associate Editor:
Comments to the Author:
(There are no comments.)

RSC Subject Editor:
Comments to the Author:
(There are no comments.)

Reviewer(s)' Comments to Author:
Reviewer: 2

Comments to the Author(s)
The authors have addressed all the questions. The quality has been improved.

Reviewer: 1

Comments to the Author(s)
All my concerns have been addressed.

Appendix A

Dear Editor and Reviewers:

Thank you for your letter and for the reviewers' comments concerning our manuscript entitled "Microbial oil production from solid-state fermentation by newly isolated oleaginous fungi from mulberry branches" (No.: RSOS-180551). Those comments are all valuable and very helpful for improving our paper, and they are also an important reference to our future researches. We have carefully considered the comments and have made corrections. Revisions are marked in the revised version.

Reviewer # 1:

Comments to authors:

In this work, the authors isolated an oleaginous fungus, *Mucor circinelloides* Q531 and studied its lipid-producing ability and the composition of the lipid. However, the written English of this manuscript should be promoted. Most importantly, the novelty of this manuscript is insufficient. So many previous have screened several microbes with lipid-producing ability and the lipid produced by *M. circinelloides* Q531 is less than previous work (reference 51).

Response: We appreciate the reviewer's positive comments.

We have polished the written English of our manuscript.

Mucor sp. is a natural fungal species, which can be used to

produce phytase in SSF processes and has been successfully employed in the production of lipids in SmF processes (Vicente et al., 2009; Roopesh et al., 2006). However, researches of *Mucor* sp. on producing microbial lipid by SSF have not been reported, moreover, lipid productivity of other species by SSF of biomass was still low (Peng et al., 2007; Immelman et al., 1997). In this study, we investigated the feasibility of direct bioconversion of mulberry branches into microbial lipids by using a newly isolated filamentous fungus that has been identified as *Mucor circinelloides* Q531. The results confirmed the lipid productivity of this strain was higher than that of the literatures because the strain was fitted better for SSF to produce lipid using Mulberry branches as raw material compared with other species and biomass.

- 1. Comment:** The authors described that ‘lipid yield decreased dramatically after d 6’ in page 4, however, additional significance test can make this result being more persuasive.

Response: We have added the significance test of the data, and revised the fig.3.

- 2. Comment:** Check throughout the manuscript and figures, all of the species names should be italic, and abbreviation of species name firstly occurred should be avoided.

Response: We have checked throughout the manuscript and figures, and revised the font of species names. The abbreviation of species name firstly occurred has been defined.

Reviewer # 2:

Comments to the Author(s):

The manuscript entitled " Microbial oil production from solid-state fermentation by newly isolated oleaginous fungi from mulberry branches" provided a newly isolated fungi that could convert mulberry branches to valuable product oil. The biomass and oil yield with cellulase activity, composition of the substrate and fatty acids were illustrated along with fermentation time. This paper need be carefully revised before being accepted in Royal Society Open Science.

Response: We appreciate the reviewer's positive comments. We have revised our manuscript carefully.

1.Comment: Since the peak value of the liquid was achieved at d 6. Authors should focus the discussion during d 0 to d 6. From Fig 5, slightly decrease of cellulose (from 45% to ~43% at d 6) could be observed (estimate p value >0.05 , if do ANOVA). Lignin almost unchanged (~37%) and Hemicellulose changed from 32% to 31% (d 6). However, yield of liquid increased significantly from 5-45 mg/gds. What is the real substrate for producing oil considering the carbon mass balance? The authors are suggested to provide the carbon mass balance analysis.

Response: The data in Fig.5 just showed changes of the proportion of lignin, cellulose and hemicellulose during SSF.

Although there were no changes significantly, the loss of lignocellulose was obvious during SSF(see Table.S1). The calculation of carbon mass balance was very complex, and we need to obtain more extra data to calculate it accurately. However, we still estimated the carbon mass balance according to our existing data. The loss of lignocellulose was 12.35% (370.5mg) after 6 days of SSF, among which the loss of cellulose and hemicellulose was about 218.3mg and 149.5mg, respectively. In addition to cell growth and residue sugar, the lost carbon (about 90mg) was used to produce lipid, which is balance with the increase of carbon (87.75mg) in lipid during SSF.

- 2. Comment:** One page 4 line 37, since the authors use “significant” to express the decrease, ANOVA should be provided.

Response: We are so sorry for the mistake of the expression “significant”. As shown in Fig.5, the content of cellulose, in comparison to hemicellulose and lignin, was decreased, but not so significant. So, we have deleted “significant”.

- 3. Comment:** Suggestion for Title: add the name of the fungi *Mucor circinelloides* to the title.

Response: We have added the name of the fungi *Mucor circinelloides* Q531 to the title.

- 4. Comment:** Suggestions for Summary: detail the time to obtain the GC analysis in line27: like at what day achieved palmitic acid (C16:0, 18.42%), palmitoleic acid (C16:1, 5.56%), stearic acid

(C18:0, 5.87%), oleic acid (C18:1, 33.89%), et al. rephrase the sentence in line30 “Meanwhile, the oleaginous fungi have a high enzyme activity of...” Language and expressions should be polished carefully, rephrase the sentence in line23 : “The a high yield...reached by...”,

Response: We have detailed the time (add “after 2 d of SSF”)to obtain the GC analysis in line 27 of Summary. We have rephrase the sentence in line30 “Meanwhile, the oleaginous fungi have a high enzyme activity of...” . The revised sentence is “Meanwhile, the oleaginous fungi have a high cellulase activity of 1.39 ± 0.09 FPU/gds”. And, we have rephrased the sentence in line23 : “The a high yield...reached by...”. The revised sentence is “The highest yield and the maximum lipid content produced by the fungal cells were 42.43 ± 4.01 mg/gram dry substrate (gds) and $28.8 \pm 2.85\%$, respectively”.

5. **Comment:** Introduction part: the authors are suggested to emphasize the key innovative point in the introduction part, but the current manuscript is just a plain description of SSF and natural fungal species. What problems did the authors solve? And Why this is interesting to the readership? Like newly isolated? Why the productivity of this strain is higher than the literature?

Response: We have revised the introduction part to emphasize the key innovation point of our manuscript. *Mucor* sp. is a natural fungal species, which can be used to produce phytase in SSF processes and has been successfully employed in the production of lipids in SmF processes (Vicente et al., 2009; Roopesh et al., 2006).

However, researches of *Mucor* sp. on producing microbial lipid by SSF have not been reported, moreover, lipid productivity of other species by SSF of biomass was still low (Peng et al., 2007; Immelman et al., 1997). In this study, we investigated the feasibility of direct bioconversion of mulberry branches into microbial lipids by using a newly isolated filamentous fungus that has been identified as *Mucor circinelloides* Q531. The results confirmed the lipid productivity of this strain was higher than that of the literatures because the strain was fitted better for SSF to produce lipid using Mulberry branches as raw material compared with other species and biomass.

6. Comment: Detail the screening process in chapter 3.2 from Materials and Methods, like what are the nine types? And what is the lipid content of the selected fungi from others?

What is the total substrate for solid-state fermentation in 3.3 chapter? Also add one sentence that indicate which fermentative indexes assayed, including liquid yield, cellulase activity, fungal biomass, et al.” Combine 3.5, 3.6 and 3.7 as one chapter.

Response: We have detailed the screening process in chapter 3.2 from Materials and Methods. In this study, we isolated nine lipid-producing fungal strains, and we found that Q531 had the highest lipid content among these strains, so we focused on the study of Q531. In our revised manuscript we have added the lipid content data of all the selected fungi (see Fig.S1).

The total substrate for SSF in chapter 3.3 was 3g (has described

in the manuscript). We have added the sentence: “Lipid yield, cellulase activity, fungal biomass, residual sugars, component of biomass and fatty acid composition were detected at different time during SSF” at the end of part 3.3.

We have combined 3.4, 3.5,3.6,3.7,3.8 and 3.9 as one chapter.

7. Comment: Integrate Fig. 3 and Fig 4 as one Figure, which could be referred to the Fig.2 from one literature (Waste Management, 2018, 76, 414-422). Be uniform as mentioned in Table 1 and Table 2 that the fermentation time (days in Fig.3 to Fig. 6) should be changed to (d).

Response: We have integrated Fig.3 and Fig.4 as one figure (Fig.3) according to the reviewer’s comment. The units of Fig.3 to Fig.6 have been uniformed according to the reviewer’s comment.

8. Comment: The GC data in line 55 on page 4 should be exhibited with the fermentation time. When did the compositions obtained?

Response: The GC data showed the compositions at different fermentation time. The compositions were obtained after 2 d of SSF. And we have added the time (after 2 d of SSF) in the manuscript.

Reviewer: #3

Comments to the Author(s):

An isolated strain can directly use mulberry branches as materials to produce microbial lipid, which is an interesting works

and dramatically saves the cost of fermentation. Besides, the writing of the manuscript is good. Attention, I think that the fourth suggestion is very important, you should make a serious consideration.

Response: We appreciate the reviewer's positive comments. We have revised our manuscript carefully.

1. Comment: For the part of introduction, line 3 to 10, page 2, please supplement more information so that readers can understand how to resolve the shortcomings of SmF by SSF.

Response: We have revised the part of introduction, and added some information and references of resolving the shortcomings of SmF by SSF.

2. Comment: Line 47-49, page 2, the content can't be read. Revise it.

Response: We have revised the sentence. The revised sentence is: "Three grams of dried lignocellulosic materials and 6 mL of mineral salt solution (MS solution contained $(\text{NH}_4)_2\text{SO}_4$, 1.7 g; KH_2PO_4 , 2.0 g; $\text{MgSO}_4 \cdot 7\text{H}_2\text{O}$, 0.5 g; $\text{CaCl}_2 \cdot 2\text{H}_2\text{O}$, 0.2 g; $\text{FeSO}_4 \cdot 7\text{H}_2\text{O}$, 0.01 g; $\text{ZnSO}_4 \cdot 7\text{H}_2\text{O}$, 0.01 g; $\text{MnSO}_4 \cdot 4\text{H}_2\text{O}$, 0.001 g; $\text{CuSO}_4 \cdot 5\text{H}_2\text{O}$, 0.0005 g; 0.1% Tween-80 (w/v), pH adjusted to 6.5) were well mixed and added to culture dishes".

3. Comment: Part 4.2, no information showing the change of lignocellulose was observed. Please check it.

Response: Fig.3 did not show the change of lignocellulose, so we delete "lignocellulose" in the sentence. The change of lignocellulose was shown in Fig.5.

4. Comment: Part 4.3, Cellulase activity has indirect effect on the lipid yield. The change in sugars concentration (glucose, maltose, etc.) during the fermentation is better beneficial to interpret the increase or decline of lipid yield. So, supplement the experimental data.

Response: We have added the data of sugars, including cellobiose, glucose, xylose and arabinose (see Fig.4).